# Physiological Response of an Oil-Producing Microalgal Strain to Salinity and Light Stress

**DOI:** 10.3390/foods11020215

**Published:** 2022-01-13

**Authors:** Zhihao Ju, Tingting Feng, Jia Feng, Junping Lv, Shulian Xie, Qi Liu

**Affiliations:** Shanxi Key Laboratory for Research and Development of Regional Plants, School of Life Science, Shanxi University, Taiyuan 030006, China; jzh0218@163.com (Z.J.); 17735831613@163.com (T.F.); fengj@sxu.edu.cn (J.F.); lvjunping024@sxu.edu.cn (J.L.); xiesl@sxu.edu.cn (S.X.)

**Keywords:** *Cyclotella menegheniana*, lipids, fatty acids, optimized culture

## Abstract

By separating and extracting algae from the collected water samples, an oil-producing diatom strain was obtained. Microscopic observation of the strain revealed that its morphological characteristics were highly similar to those of the genus *Cyclotella*. The cloning of 18S rDNA and phylogenetic analysis showed that the algae were clustered with *Cyclotella menegheniana* with a high support rate, indicating that the alga was *C. menegheniana*. The fatty acid content of the alga was determined and found to be mainly C14, C16, and C18 fatty acids, which were in accordance with the relevant standards for edible oil. In this study, different gradient levels of salinity and light were set to investigate the culture and bioactive substance production of *C. menegheniana*. The results showed that the best growth condition was achieved when the salinity was 15 g·L^−1^, and its biomass and oil content were the highest at 0.27 g·L^−1^ and 21%, respectively. The final biomass was the highest when the light intensity was 2000 Lux and the oil content was 18.7%. The results of the study provided a basis for the large-scale production of edible oils and biodiesel.

## 1. Introduction

Microalgae are a resource of great exploitation and use, with about one million species of algae currently distributed worldwide, including more than 50,000 oil-producing algal species [1]. Microalgae can fix carbon dioxide through photosynthesis, convert light energy into chemical energy, and store it in the form of oil in the cells. In addition, microalgae contain proteins, carbohydrates, and various minerals, which are important sources for the production of food, pharmaceuticals, and biodiesel [2].

With the excessive population growth, traditional edible fats and oils are no longer able to meet the needs of food and industry. Relevant studies have shown that microalgal oil can be used as an alternative to edible fats and oils [3]. Microalgal oil is a single-cell oil, and the oil content is related to the microalgal species and culture conditions in which the fatty acids are composed of C14–C20 long-chain fatty acids and triglycerides, with C16 and C18 series fatty acids predominating. The oil content and fatty acid composition of microalgal oil are similar to those of vegetable oil; therefore, it can be developed and used as edible oil.

Nowadays, the environmental pollution caused by the consumption of petroleum fuels is becoming extremely serious, and research on biodiesel has been initiated. Compared with regular diesel, biodiesel contains the correct cetane number and a higher oxygen content with better combustion performance and a lower engine emission rate [4]. Biodiesel is a clean energy source that has physical and chemical properties similar to petroleum diesel and can be used directly in conventional engines. In addition, it is renewable, nontoxic, and degradable and does not cause a net accumulation of greenhouse gases. Therefore, conducting research on biodiesel is of great practical importance.

Thus, the selection and breeding of suitable algal species are key issues for developing microalgal energy. In this study, the algal strain MH2018001 was judged as a strain of *C. menegheniana* by microscopic observation and phylogenetic analysis, and it was cultured under different salinity and light conditions to analyze the optimal culture conditions and the maximum yield of oil and grease for this strain, so as to lay the foundation for further development of microalgal oil and improve the productivity of its edible oil and grease.

## 2. Materials and Methods

### 2.1. Sample Collection and Separation

The water samples were collected from the Python River Scenic Area in Shanxi Province, preserved in sterile centrifuge tubes, added to sterile deionized water, and shaken overnight in a shaker. Single colonies were screened by the plate isolation method [5], and this strain was inoculated into 250 mL triangular flasks in an ultra-clean bench and kept in the culture room.

### 2.2. Algal Species Culture and Identification

#### 2.2.1. Algal Species Culture

The isolated algal strains were expanded using the D1 medium, which was prepared as shown in Table 1. The cultivation light color was cool white, light intensity in the culture room was 2000 Lux, the light–dark cycle ratio was 12 h:12 h, and the incubation temperature was 25 °C.

#### 2.2.2. Observation of Algal Cell Morphology and Detection of Oil-Producing Algal Strain

The algal cell morphology was observed under an Olympus BX-51 microscope (Olympus Corp, Tokyo, Japan) equipped with a microphotographic digital camera (DP72) after taking an appropriate amount of algal solution and fixing it with Ruger’s reagent [6].

An appropriate amount of algal solution in the stable growth phase was centrifuged, precipitated, and transferred to a 96-well black enzyme standard plate, and the assay was performed according to the literature [7]. A small amount of sample mount was removed, and the lipid droplet morphology was observed under a fluorescence microscope (Olympus TL4, Tokyo, Japan) [8].

#### 2.2.3. Algal Strain 18S rRNA Gene Amplification and Sequencing

The genomic DNA was extracted using the SDS method [9].

The 18S rDNA amplification forward primer G01: 5′-CACCTGGTTGATCCTGCCAG-3′ and reverse primer G07: 5′-AGCTTGATCCTTCTCTGCAGGTTCACCTAC-3′ (synthesized by Shanghai Sangon Biotech Co., Ltd., Shanghai, China) were used. The polymerase chain reaction (PCR) system (20 μL) contained 11.8 μL of ddH_2_O, 2 μL of DNA buffer, 2 μL of dNTPs, 1.5 μL of primer F, 1.5 μL of primer R, 1 μL of template DNA, and 0.2 μL of Taq DNA polymerase. The PCR reaction procedure was as follows: 95 °C predenaturation for 5 min; 94 °C denaturation for 1 min 50 s, 54 °C annealing for 45 s; 72 °C for 1 min, 72 °C extension for 10 min, for 30 cycles; and storage at 4 °C.

Electrophoresis was performed using 1.0% agarose gel (DYY-7C, Beijing Liuyi Instrument Factory, Beijing, China), and the products were recovered and sent for testing (Beijing Genomics Institution, Beijing, China). The MEGA software was used to construct neighbor-joining and maximum likelihood trees [10,11] and MrBayes 3.1.2 [12] was used to construct Bayesian inference trees.

### 2.3. Determination of Physiological and Biochemical Indicators of Algal Plants

#### 2.3.1. Determination of the Fatty Acid Content

The extracted algal oil was first methylated by dissolving the dried algal oil in chloroform. Then, it was transferred to a 1.5 mL Agilent bottle, 1 mL of 1 μmol m^−2^ s^−1^ methanolic sulfate solution was added, and nitrogen was filled and sealed. A 100 °C water bath was heated for 1 h. After cooling, 200 μL of ultrapure water was added and shaken to mix well. The organic phase was extracted three times by allowing it to stand with 200 μL of hexane, combined and transferred to another clean 1.5 mL Agillient bottle, blow-dried with nitrogen, and left to be measured [13].

The fatty acid content was determined using gas chromatography–mass spectrometry (7890A-5975C, Agilent, LA, USA) [14]. The molecular structure of each component was determined, and finally the relative content was calculated using the peak area normalization method [15].

#### 2.3.2. Determination of End Biomass

The method proposed by Lv et al. [16] was used with some modifications. The biomass dry weight of the algal solution was determined by baking a 0.45 μm microporous filter membrane at a constant temperature and weighing it (*m*_1_), pumping the algal solution with a vacuum pump, and baking it at a constant temperature and weighing it (*m*_2_); and the difference was the dry weight of the algal solution. Three replicates were set up in each group, and the results were recorded. The dry weight of algal cells was calculated using the following equation:DW = (*m*_2_ − *m*_1_)/*V*
where DW indicates the dry weight of algal cells (g·L^−1^), *m*_1_ (g) indicates the constant weight of the filter membrane, *m*_2_ (g) indicates the sum of the weight of the algae after extraction and the weight of the filter membrane, and *V* indicates the total volume of the algal solution.

#### 2.3.3. Determination of Chlorophyll Fluorescence Parameters

The parameters related to chlorophyll fluorescence were measured by a portable chlorophyll fluorescence instrument (AquaPen-C AP-C10, CZE). Three milliliters of algae solution was taken to a dark and opaque place to adapt for 30 min, and then the determination was made [17]. A one-way ANOVA test was performed with SPSS 19.0 statistical software (IBM Inc, Chicago, IL, USA).

#### 2.3.4. Determination of the Chlorophyll Content

The chlorophyll content was determined according to a previous study [18]. First, 5 mL of algal solution was centrifuged, an equal dose of 95% ethanol was added, extraction was performed for 24 h under light-proof conditions at 4 °C, and the supernatant was collected after centrifugation. The absorbance values were measured at 649 nm and 665 nm using an ultraviolet–visible spectrophotometer. The intra-algal chlorophyll a content was calculated using the following equation:Chlorophyll a = 13.95 × A665 − 6.88 × A649

#### 2.3.5. Determination of the Total Lipid Content

The extraction method for total lipids from a previous study was followed with some modifications [19]. A certain amount of freeze-dried algal powder (*W*_1_) was weighed using an analytical balance (TB-214, BSISL, Beijing, China), and placed in a glass vial. Two milliliters of chloroform–methanol (2:1) was added, then crushed and centrifuged, and the supernatant was transferred to the weighed glass vial (*W*_2_). The remaining algal residue was again mixed with the chloroform–methanol solution and centrifuged until the algae turned white. All liquids were combined in 5-mL glass vials and blow-dried with nitrogen and weighed (*W*_3_, g) [20]; three replicates were set in each group. The total lipid content was calculated as follows:LC (%DW) = (*W*_3_ − *W*_2_)/*W*_1_ × 100%

#### 2.3.6. Statistical Analysis

Three replicates were set up for each group of experiments, and all measured values were expressed as mean standard deviations and analyzed for significance using one-way ANOVA in SPSS 19.0 (IBM Inc, Chicago, IL, USA) combined with the LSD method (Least Significant Difference method), when *p* < 0.05 indicated a statistically significant difference.

## 3. Results and Analysis

### 3.1. Morphological Identification and Phylogenetic Analysis of Algal Strains

The results were observed using a light microscope and scanning electron microscope, as shown in Figure 1 and Figure 2. In LM images, valves are disc-shaped. In SEM images, the valve faces transversely undulate. The central area is distinct and isolated from the marginal chambered striae. The central area covers 1/3 of the valve face. Based on morphological observations, the strain MH2018001 was similar to *Cyclotella menegheniana* [21]. After staining with Nile Red, observation using fluorescence microscopy revealed a clear bright yellow or orange fluorescence in the cells, indicating the presence of oil components in this strain.

The 18S rDNA sequence was obtained by amplifying and sequencing the genomic DNA of the algal strain MH2018001 as a template. The results showed that the strain was highly similar to *C. menegheniana*. A phylogenetic tree based on the 18S rDNA sequence in Figure 3 shows that the strain was clustered with *C. menegheniana*, and the support rate was also high.

### 3.2. Fatty Acid Composition Analysis

The fatty acid content of *C. menegheniana* MH2018001 is shown in Table 2, in which eight fatty acids were detected, of which C16 and C20 were the main components, with a total content of 80.7%. Reports have shown that the C16 and C18 families are common feedstocks for biodiesel production [22], whereas the C20 family comprises highly unsaturated fatty acids essential for humans and animals and plays a key role in growth and development [23].

The major fatty acids in *C. menegheniana* MH2018001 were found to be palmitic acid (C16:0), palmitoleic acid (C16:1), and eicosapentaenoic acid (C20:5), with contents of 20.79%, 27.59%, and 19.46%, respectively. In addition, saturated fatty acids and monounsaturated fatty acids (MUFA) also accounted for a large proportion, at 27.7% and 32.43%, respectively.

### 3.3. Analysis of Algal Biomass under Different Salinities and Light Levels

Figure 4A shows that the initial dry weight of the three salinities was about 0.04 g·L^−1^. In 0–2 days, the dry weight of *C. menegheniana* MH2018001 reached 0.11 g·L^−1^. *C. menegheniana* MH2018001 entered the logarithmic phase, and the growth rate accelerated and the dry weight increased. The dry weight values reached stability at between 8 and 12 days, and the growth of *C. menegheniana* MH2018001 entered a stable phase, reaching up to 0.28 g·L^−1^ after 12 days. This indicated that too high a salinity is not conducive to biomass accumulation.

Figure 4B shows that the initial dry weight of the three photons was 0.05 g·L^−1^, and the algal plants grew rapidly from 0 to 2 days, reaching a dry weight of 0.11 g·L^−1^. After 6 days, the photometric 4000 Lux final biomass increased significantly, becoming higher than the samples at 2000 Lux and 6000 Lux, and reached a maximum biomass of 0.27 g·L^−1^ after 14 days. The results indicated that the highest final biomass was obtained at 4000 Lux.

### 3.4. Analysis of Chlorophyll Fluorescence Parameters of Algal Biomass under Different Salinity and Light Conditions

Figure 5A,B show that the changing trends of fluorescence parameters Fv/Fm and Fv/Fo are essentially the same for different salinity groups. From 0 to 4 days, the chlorophyll fluorescence parameters for each group of salinities began to differ, but the overall trend was increasing, and these parameters were slightly higher for the sample at a salinity of 15 g·L^−1^ than for the samples at other salinities. After 4 days of incubation, the chlorophyll fluorescence parameters of the three groups began to show a decreasing trend. After 10 days, the chlorophyll fluorescence parameters reached the maximum value at the salinity of 15 g·L^−1^. After 10 days, the chlorophyll fluorescence parameters reached the maximum value of 15 g·L^−1^. In the early stage of microalgae culture, the intracellular nutrients were more abundant, which made the chlorophyll fluorescence parameters show an increasing trend, and the cells lacking nutrients showed a decreasing trend [24]. It is shown that high and low salinity affects the photosynthetic capacity and PSII activity of *C. menegheniana* MH2018001.

Figure 5C,D show that on the first day after inoculation, the chlorophyll fluorescence parameters Fv/Fm of all groups increased significantly, and the luminosity of the sample with 40 μmol m^−2^ s^−1^ was significantly higher than that of the other two groups. After four days, the gap gradually widened, and the luminosity of the samples with Fv/Fm of 80 μmol m^−2^ s^−1^ and 120 μmol m^−2^ s^−1^ was much lower than that of the sample with 40 μmol m^−2^ s^−1^. This indicates that the enhancement photometric had little effect on the photosynthetic capacity and PSII activity of *C. menegheniana* MH2018001.

### 3.5. Analysis of Chlorophyll Content of Algal Biomass under Different Salinities and Light Levels

Figure 6A shows that the content of *C. menegheniana* MH2018001 showed an increasing and then decreasing trend at all three salinities, but the differences were more pronounced, and the chlorophyll content did not change significantly at a salinity of 25 g·L^−1^. From 0 to 5 days, the chlorophyll content was higher at a salinity of 15 g·L^−1^ than at 5 g·L^−1^. After 5 days, the chlorophyll content was significantly higher at a salinity of 5 g·L^−1^ than at a salinity of 15 g·L^−1^ (*p* < 0.05). Conversely, after 5 days, the salinity of 5 g·L^−1^ was significantly higher than 15 g·L^−1^ (*p* < 0.05) and reached the highest value of 2.6 mg·L^−1^. The results indicated better photosynthetic activity at low-salinity incubation, which contributed to the accumulation of chlorophyll a content. However, chlorophyll a accumulation was slow and at a low level in the later stages of culture.

Figure 6B shows that the chlorophyll a content of *C. menegheniana* MH2018001 at 2000 Lux was significantly higher than that of the rest of the groups, and the chlorophyll a content showed an increasing and then decreasing trend in all three conditions, with the highest content after 2–4 days. The chlorophyll a content of *C. menegheniana* MH2018001 under the three conditions was significantly different from 4 days onward. Throughout the cycle, the chlorophyll a content of the sample at 2000 Lux was clearly higher than that of the other two groups, and significantly higher than the samples at 4000 Lux and 6000 Lux after 4 days (*p* < 0.05). Thus, low luminosity was more favorable for chlorophyll a synthesis.

### 3.6. Analysis of Total Lipid Content of Algal Biomass under Different Salinities and Light Levels

Figure 7A shows an overall increasing trend in the oil content of *C. menegheniana* MH2018001 with increasing salinity. The highest lipid content (21%) was reached at 25 g·L^−1^. However, at this salinity, the growth rate of *C. menegheniana* MH2018001 was slow and the biomass obtained was low, whereas at a salinity of 15 g·L^−1^, the biomass was 1.3 times higher than that at 25 g·L^−1^. Therefore, a salinity of 15 g·L^−1^ is the optimum culture concentration for *C. menegheniana* MH2018001.

Figure 7B shows that the light intensity could affect the lipid content of *C. menegheniana* MH2018001, which showed an increasing, then decreasing trend, which then increased with the increase in light intensity. The highest oil content was 18.7% at 2000 Lux, followed by that at 6000 Lux, which was significantly higher than that at 4000 Lux (*p* < 0.05). Since the sample at 2000 Lux had the highest biomass and higher lipid content, the optimal light intensity for culture of *C. menegheniana* MH2018001 was determined to be 2000 Lux.

## 4. Discussion

The morphology of algal strain MH2018001 was highly similar to that of *Cyclotella menegheniana* as observed using light microscopy and scanning electron microscopy. The cloning of the 18S rDNA and the construction of a phylogenetic tree showed that MH2018001 was a *C. menegheniana* strain.

In analyzing fatty acid composition, eicosanoids, such as eicosapentaenoic acid (C20:5), are ω-3 polyunsaturated fatty acids and an integral part of a healthy diet. The cetane number is a key indicator of biodiesel and can affect the combustion and emissions of biodiesel [25]. In this study, this strain of *C. menegheniana* MH2018001 was found to contain both polyunsaturated fatty acids and MUFA, the carbon chain length was 14–20 carbons, and the unsaturated double bonds did not exceed four. Li [26] showed that the C16, C18, and C20 series were the major fatty acids by analyzing the fatty acids of three diatoms, which was slightly different from the results of the present experimental study, and which may be related to the different algal species or the different culture conditions used.

The chlorophyll content can reflect the growth of algal cells in a specific environment and is the main indicator of algal biomass in the water column and an important parameter for eutrophication evaluation in lakes. Valenzuela-Espinoza [27] explored the relationship between *f*/2 medium lipids and substances, such as chlorophyll, and showed that the chlorophyll content values positively correlated with the algal cell growth rate. This was consistent with the results of the present study, where chlorophyll a increased with time and reached a stable phase after nutrient depletion, cells gradually died, algal cell density decreased, growth rate decreased, and chlorophyll a content began to decrease again.

Under appropriate salt stress, the growth conditions of microalgae can be altered, which, in turn, affects their lipid content. Pahl [28] found that the total lipid content of *Microcystis aeruginosa* was strongly influenced by salinity, and the maximum lipid content was reached when the salinity was 11.2 psu, which was consistent with the results of this experiment. The results of the present experimental study found that algal cells grew more slowly when the salinity was 0.01 g·L^−1^, because too low a salinity affected their cellular osmotic pressure, affecting the ability of algal cells to absorb nutrient salts and therefore also affecting lipid production. Meridith [29] found that marine microalgae grew best and had the highest biomass at a salinity of 22 psu, and when the salinity increased in the later stages of culture, the oil content increased significantly. This study showed that high-salinity conditions (salinities higher than 15) inhibited the growth of algae, but high salinity favored the accumulation of total algal lipids; however, the differences in the optimal salinity and chlorophyll trends might vary from species to species. Adams [30] showed that suitable salinity could increase the accumulation of lipids in *C. menegheniana*, which was consistent with the results of this experimental study. Therefore, suitable salinity should be selected to improve lipid production.

As the main source of energy in the growth and development of algae, the intensity of light has even more important effects on the growth of microalgae and changes in their biochemical composition. Cheirsilp [31] found that an increase in light intensity decreased the lipid content of some species, but promoted or did not affect the lipid production of other species [32]. Ji [33] found that the oil content of *C. menegheniana* reached 56%; however, the oil content in this experimental study was 21%, which might have been caused by differences between growth environments and different strains of algae used and hence the high and low oil content. In this study, the low light level was favorable for the accumulation of total lipids in *C. menegheniana*, and the algae had higher total lipid content under a 2000 Lux light intensity, which was consistent with the studies of Chen [34] and Wahidin [35], where an appropriate light level promoted lipid accumulation and increased the lipid yield.

## 5. Conclusions

The algal strain MH2018001 collected from the scenic area of the Python River in Shanxi Province was identified by microscopic observation and 18S rDNA gene sequence comparison as a strain of *C. menegheniana*. The eicosapentaenoic acid (C20:5) was found to have more health and medical applications through the determination of fatty acids. A large amount of lipid distribution in the cells could be clearly seen after Nile Red staining. The lipid yield could reach 21% with a biomass of 0.28 g·L^−1^ after 14 days of incubation by salt stress, and 18.6% by changing the light intensity. This study showed that the strain of *C. menegheniana* had a high growth rate and high oil yield, which was appropriate for industrial application.

## Figures and Tables

**Figure 1 foods-11-00215-f001:**
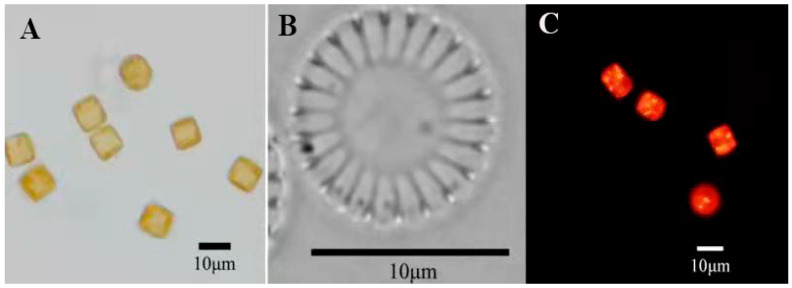
*Cyclotella**menegheniana* MH2018001 ((**A**,**B**) light micrographs; (**C**) fluorescence micrographs).

**Figure 2 foods-11-00215-f002:**
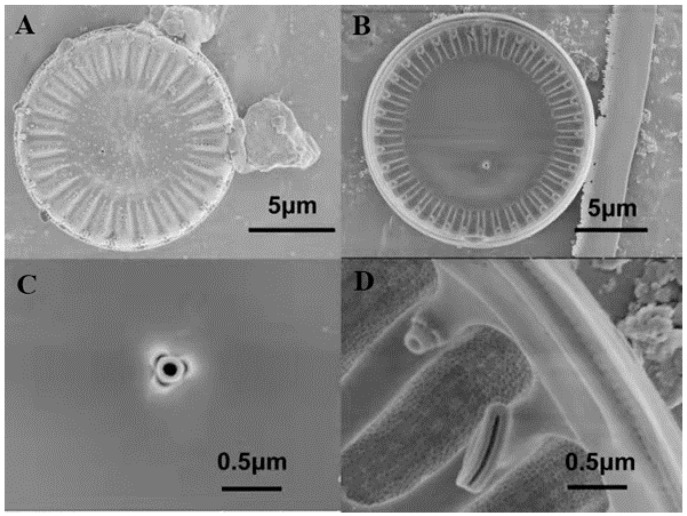
SEM micrographs of *Cyclotella menegheniana* MH2018001. (**A**) external view of *C. menegheniana* MH2018001; (**B**) internal view of *C. menegheniana* MH2018001; (**C**) showing the fultoportula; (**D**) showing the rimoportula positioned on a mantle costa).

**Figure 3 foods-11-00215-f003:**
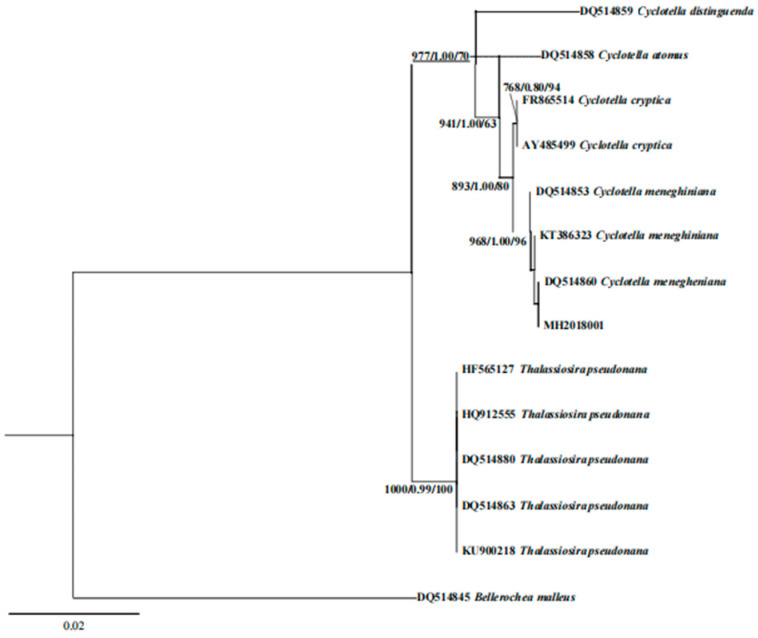
Bayesian analysis tree based on 18S rDNA gene sequence of *Cyclotella*
*menegheniana* MH2018001. The numerical values at the nodes represent the support values of BI bootstrap/ML bootstrap/NJ bootstrap.

**Figure 4 foods-11-00215-f004:**
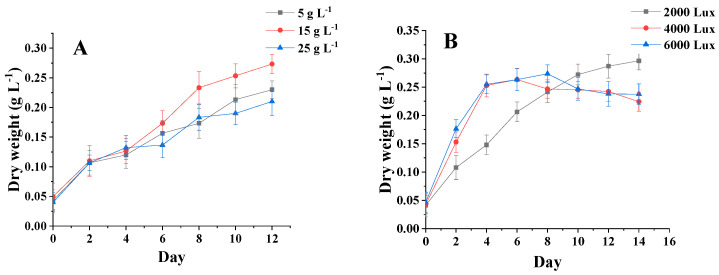
Biomass under different culture conditions of *C. menegheniana* MH2018001. (**A**) represents the biomass at different NaCl concentrations; (**B**) represents the biomass under different light conditions). The results are presented as mean ± SE (*n* = 3).

**Figure 5 foods-11-00215-f005:**
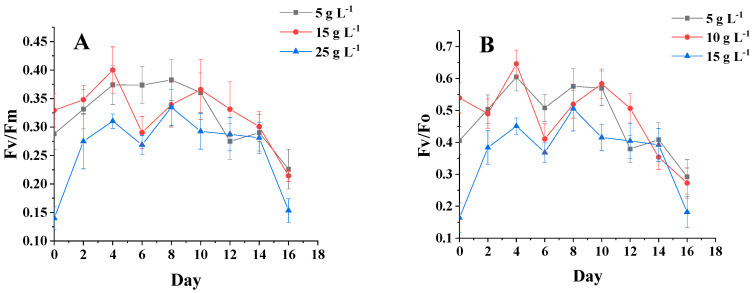
Chlorophyll fluorescence parameters under different culture conditions. (**A**,**B**) represents Fv/Fm and Fv/Fo at different NaCl concentrations; (**C**,**D**) represents Fv/Fm and Fv/Fo at different light conditions). The results are presented as mean ± SE (*n* = 3).

**Figure 6 foods-11-00215-f006:**
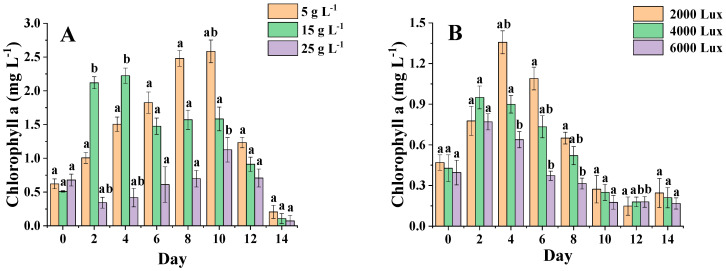
Chlorophyll content under different culture conditions. (**A**) represents the chlorophyll content at different NaCl concentrations; (**B**) represents the chlorophyll content under different light conditions). Different letters indicate statistical difference between groups (*p* < 0.05). The results are presented as mean ± SE (*n* = 3).

**Figure 7 foods-11-00215-f007:**
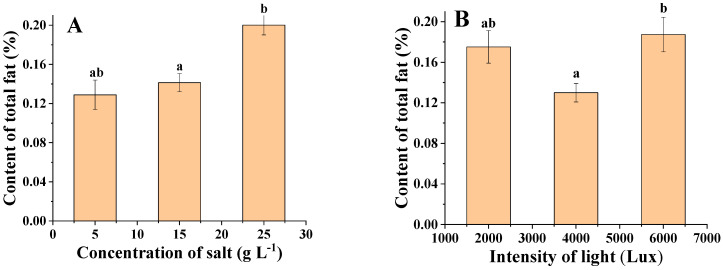
Triglyceride contents under different culture conditions of *C. menegheniana* MH2018001. (**A**) represents the oil content at different NaCl concentrations; (**B**) represents the oil content under different light conditions). Different letters indicate statistical difference between groups (*p* < 0.05). The results are presented as mean ± SE (*n* = 3).

**Table 1 foods-11-00215-t001:** Medium components of D1.

Components	Dosage
NaNO_3_	0.12 g
K_2_HPO_4_	0.04 g
MgSO_4_·7H_2_O	0.07 g
KH_2_PO_4_	0.08 g
CaCl_2_·2H_2_O	0.02 g
NaSiO_3_·9H_2_O	0.1 g
NaCl	0.01 g
MnSO_4_·4H_2_O	0.1 mL
Ferric citract	0.005 g
Soil extract	20 mL
A5 solution	1 mL
Distilled water	979 mL
NaNO_3_	0.12 g
K_2_HPO_4_	0.04 g
MgSO_4_·7H_2_O	0.07 g

**Table 2 foods-11-00215-t002:** Fatty acid compositions of *C.*
*menegheniana* MH2018001.

Item	Chemical Formula	Content
C14:0	C_14_H_28_O_2_	7.24%
C15:0	C_15_H_30_O_2_	0.37%
C16:0	C_16_H_36_O_2_	20.79%
cis C16:1	C_16_H_30_O_2_	27.59%
cis C16:2	C_16_H_28_O_2_	2.08%
cis C16:3	C_16_H_26_O_2_	10.78%
cis C18:1	C_18_H_34_O_2_	5.3%
cis C20:5	C_20_H_30_O_2_	19.46%
Others	-	6.39%
SFA	-	27.7%
MUFA	-	32.43%
PUFA	-	32.1%

Note: “-” means none.

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
