# Peer review of "Physiological Response of an Oil-Producing Microalgal Strain to Salinity and Light Stress"

_foods, 2022, doi:10.3390/foods11020215_

Round 1

Reviewer 1 Report

I read the paper “Physiological response of an oil-producing microalgal strain to  salinity and light stress”, which is aimed to explore the effects of  different gradient levels of salinity and light on bioactive substance production such as oil and grease from . The results were examined, to ascertain increases in edible products from C. menegheniana strain. The authors suggested that the strain  of C. menegheniana had a high growth rate and high oil yield, which was appropriate for industrial application.

This study shows points of interest. However, it has some criticisms, which should be corrected.

  1. TAGs play a fundamental role in energy storage within the microalgae cell, in the results changes of total fat have been reported. Are the values ​​reported as TGAS superimposable to those of fat content?
  2. Previous results the color of the cultivation light affects both the total lipid content and the fatty acid profile. A sentence to experimental conditions should be added

       3. In the legends of figures, the statistical criteria must be reported.

Author Response

  1. Q: TAGs play a fundamental role in energy storage within the microalgae cell, in the results changes of total fat have been reported. Are the values ​​reported as TGAS superimposable to those of fat content?

A: Yes, some studies have shown that TAGs play a fundamental role in energy storage and total fat changes in microalgae cells. In this study, we did not analyze the values of TAGs, and we only analyzed fatty acids content, which are components and precursors of TAGs.

  1. Q: Previous results the color of the cultivation light affects both the total lipid content and the fatty acid profile. A sentence to experimental conditions should be added.

A: We added “The cultivation light color is cool white,” in the mauscripte. Please see the highlight parts in L58.

  1. Q: In the legends of figures, the statistical criteria must be reported.

A: We added the statistical criteria. Please see the highlight parts in L229, L262, L293.

Reviewer 2 Report

Generally, the manuscript is well prepared. However, a few shortcomings need to be corrected:

Materials and Methods

  • In this section, information on the number of repetitions in which individual measurements and determinations were performed should be completed
  • The Authors did not provide methods for statistical processing of the results

Results and analysis

  • L 191-192: The Authors state that eight fatty acids were detected in the analyzed material, mainly C16 and C20. Which acids are included in the Others group (Table 2)?
  • L 201: Table 2. A valuable supplement to the data in Table 2 would be to distinguish between cis and trans acids and assign them to omega groups..
  • The Authors performed a statistical analysis of the research results and used homogeneous groups to evaluate the differences between the object means. The manuscript does not mention the test and significance level of these calculations. This information should be included in Figure 5 (L 238-239) and Figure 6 (L: 270-271).

Author Response

We have completed our revised manuscript “Physiological response of an oil-producing microalgal strain to salinity and light stress”. We appreciate the helpful review. Based on the review we have amended many of the changes suggested by the reviewer, including:

Materials and Methods

  1. Q: In this section, information on the number of repetitions in which individual measurements and determinations were performed should be completed.

The Authors did not provide methods for statistical processing of the results

   A: We add Statistical analysis in Materials and Methods. Please see highlight parts L126 - 130.

Results and analysis

  1. Q: L 191-192: The Authors state that eight fatty acids were detected in the analyzed material, mainly C16 and C20. Which acids are included in the others group (Table 2)?

A: Others acids include C19:1, C19:2, C19:3, etc. There are many kinds of them, but they each account for small amount. We put them into others group.

  1. Q: L 201: Table 2. A valuable supplement to the data in Table 2 would be to distinguish between cis and trans acids and assign them to omega groups.

   A: We marked the cis in Table 2. Please see the highlight part in Page 6.

  1. Q: The Authors performed a statistical analysis of the research results and used homogeneous groups to evaluate the differences between the object means. The manuscript does not mention the test and significance level of these calculations. This information should be included in Figure 5 (L 238-239) and Figure 6 (L: 270-271).

A: We added the statistical criteria in Figure 5, Figure 6 and Figure 7. Please see the highlight parts in L229, L262, L293.

In addition, according to Ms. Zane Zhang’s advise, we added the contents of “2.3.3. Determination of chlorophyll fluorescence parameters” and “3.4. Analysis of chlorophyll fluorescence parameters of algal plants under different salinity and light conditions” in the manuscript to support the research results. Please look at the part marked in red.

We hope you find everything in order with this revision. If you have any questions please do not hesitate to contact me.

Sincerely,

Liu Qi
